# Does Telotristat Have a Role in Preventing Carcinoid Heart Disease?

**DOI:** 10.3390/ijms25042036

**Published:** 2024-02-07

**Authors:** Aura D. Herrera-Martínez, Antonio C. Fuentes-Fayos, Rafael Sanchez-Sanchez, Antonio J. Montero, André Sarmento-Cabral, María A. Gálvez-Moreno, Manuel D. Gahete, Raúl M. Luque

**Affiliations:** 1Maimonides Institute for Biomedical Research of Cordoba (IMIBIC), 14004 Córdoba, Spain; b22fufaa@uco.es (A.C.F.-F.); patologiahrs@gmail.com (R.S.-S.); antonio.montero.96@gmail.com (A.J.M.); amsbcabral@gmail.com (A.S.-C.); mariaa.galvez.sspa@juntadeandalucia.es (M.A.G.-M.); bc2gaorm@uco.es (M.D.G.);; 2Endocrinology and Nutrition Service, Reina Sofia University Hospital, 14004 Córdoba, Spain; 3Department of Cell Biology, Physiology, and Immunology, University of Córdoba, 14004 Córdoba, Spain; 4CIBER Physiopathology of Obesity and Nutrition (CIBERobn), 14004 Córdoba, Spain; 5Reina Sofia University Hospital, 14004 Cordoba, Spain; 6Pathology Service, Reina Sofia University Hospital, 14004 Córdoba, Spain

**Keywords:** telotristat, serotonin, carcinoid heart disease

## Abstract

Carcinoid heart disease (CHD) is a frequent and life-threatening complication in patients with carcinoid tumors. Its clinical management is challenging is some cases since serotonin-induced valve fibrosis leads to heart failure. Telotristat is an inhibitor of tryptophan-hydroxylase (TPH), a key enzyme in serotonin production. Telotristat use in patients with carcinoid syndrome and uncontrollable diarrhea under somatostatin analogs is approved, but its specific role in patients with CHD is still not clear. IN this context, we aimed to explore the effect of telotristat in heart fibrosis using a mouse model of serotonin-secreting metastasized neuroendocrine neoplasm (NEN). To this aim, four treatment groups (n = 10/group) were evaluated: control, monthly octreotide, telotristat alone, and telotristat combined with octreotide. Plasma serotonin and NT-proBNP levels were determined. Heart fibrosis was histologically evaluated after 6 weeks of treatment or when an individual mouse’s condition was close to being terminal. Heart fibrosis was observed in all groups. Non-significant reductions in primary tumor growth were observed in all of the treated groups. Feces volume was increased in all groups. A non-significant decrease in feces volume was observed in the octreotide or telotristat-treated groups, while it was significantly reduced with the combined treatment at the end of the study compared with octreotide (52 g reduction; *p* < 0.01) and the control (44.5 g reduction; *p* = 0.05). Additionally, plasma NT-proBNP decreased in a non-significant, but clinically relevant, manner in the octreotide (28.2% reduction), telotristat (45.9% reduction), and the octreotide + telotristat (54.1% reduction) treatment groups. No significant changes were observed in plasma serotonin levels. A similar non-significant decrease in heart valve fibrosis was observed in the three treated groups. In conclusion, Telotristat alone and especially in combination with octreotide decreases NT-proBNP levels in a mouse model of serotonin-secreting metastasized NEN, when compared with the control and octreotide, but its effect on heart valve fibrosis (alone and in combination) was not superior to octreotide in monotherapy.

## 1. Introduction

Carcinoid syndrome (CS) is the most frequent hormonal complication in serotonin-secreting neuroendocrine neoplasms (NENs) [1,2] and is clinically characterized by the presence of chronic diarrhea and/or flushing, accompanied by systemic elevated levels of serotonin or its metabolite 5-hydroxyindolacetic acid (5-HIAA) and other symptoms, including bronchospasm or abdominal pain [1,3]. CS is predominantly observed in patients with well-differentiated NENs, with the small intestine being the most frequent primary tumor location, followed by the lung [4]. Additionally, other primary tumor locations include the pancreas, ovarium, and thymus [3,4]. Importantly, patients with CS frequently present associated fibrotic complications, especially mesenteric fibrosis and carcinoid heart disease (CHD) [5]. Specifically, CHD is observed in about 50% of patients with CS, and is considered a major source of morbidity and mortality [5].

Specifically, plaque-like fibrous deposits occur on right-side heart valves and endocardial surfaces [6]. These deposits induce tricuspid and pulmonary valve regurgitation and/or stenosis; consequently, right ventricular volume overload is produced, resulting in right heart failure [4,6]. In fact, large amounts of vasoactive substances (serotonin, tachykinins, and prostaglandins) reach the right side of the heart in these patients, which is mainly due to reduced hepatic metabolism, usually in NENs with metastatic liver involvement [6,7]. Generally, the left side of the heart is spared in these patients due to the vasoactive substance metabolization that occurs in the lungs [8].

Therefore, the presence of CHD gives CS a character of aggressiveness. Indeed, CHD can progress rapidly, and thus regular screening for CHD should be performed in these patients using echocardiograms and plasma N-terminal pro-brain natriuretic peptide (NT-proBNP) [3]. For these reasons, controlling CS with medical treatment is essential. Surgery of the primary tumor in patients with metastasis is indicated only in patients with intestinal obstructions or for reducing tumor load in specific cases [3,4]. In this context, somatostatin analogs (SSAs) are recognized as the first line of treatment for patients with CS, especially long-acting preparations of SSAs [3,4,9,10]. Despite this, in some cases, refractory CS might be present, which is characterized by recurring or persisting symptoms, while urinary 5-hydroxyindoleacetic acid (u-5HIAA) levels increase or remain elevated, despite the use of maximum-label doses of SSAs [3]. Therefore, second-line treatments should be used in these cases, which include direct hormone secretion control [3]. To this aim, telotristat ethyl, a tryptophan hydroxylase (TPH) inhibitor, is the rate-limiting step in the secretion of serotonin [11]. Previous studies have reported that its metabolite, the hippurate salt of telotristat ethyl, reduces serotonin levels through the gastro-intestinal tract in mice and improves clinical symptoms [12]. Clinical studies in CS patients not adequately controlled under SSA treatment have reported significant reductions in their number of bowel movements and in u5-HIAA levels after telotristat treatment [12,13,14]. Based on these data, telotristat is currently recommended in clinical guidelines [3,5].

Despite the fact that the role of telotristat for controlling diarrhea and flushing has been clearly reported [5], its specific effect for preventing or reversing right-valve fibrosis, and consequently CHD, is not clearly known. In this context, the aim of this study was to generate a mouse model of CHD to evaluate the effect of telotristat ethyl on right-valve fibrosis, and compare its effect not only with a placebo, but also with octreotide, which is the first line of treatment for controlling CS symptoms. Additionally, a combined treatment of telotristat and octreotide was evaluated, since the use of telotristat in clinical practices is currently approved in combination with SSAs.

## 2. Results

A xenograft model of serotonin-secreting NEN was established wherein primary tumors and metastases were observed at sacrifice (Figure 1). Valve fibrosis was also observed in all groups according to the histological analysis.

When each primary tumor was evaluated, slight and non-significant reductions in tumor growth were observed in the octreotide group (11.05%), the telotristat group (23.6%), and the combination group (15.64%) (Figure 2).

As is expected in a CS model, the feces volume significantly increased at the end of the study in both the control and the treatment groups (octreotide, telotristat, or their combination; Figure 3A). Specifically, feces volume (determined at baseline and the week before sacrifice) in the control group (73.1 ± 19.2 g) was slightly higher than in the telotristat-treated group (67.51± 2.54 g, *p* = 0.09), and significantly higher than in the combination-treated group (28.6 ± 2.7 g, *p* = 0.03) (Figure 3B). This volume was slightly increased in the octreotide group (80.6 ± 9.8 g) when compared with the telotristat group (*p* = 0.05), the combination group (*p*= 0.006), and even with the control group in a non-significant manner (*p* = 0.31) (Figure 2B).

After tumor development and treatment, the mice did not exhibit significant changes in their total body weight across the different groups (Figure 4A). Moreover, lean mass was also similar in all groups (25.32 ± 3.28 in the placebo group; 22.90 ± 2.93 in the octreotide group; 23.93 ± 3.49 in the telotristat group; and 23.02 ± 3.89 in the combination group) (Figure 4B), while fat mass slightly decreased, in a non-significant manner, in all drug-treated groups compared with the control group [25.32 g ± 3.28 g vs. 22.9 g ± 2.93 in the octreotide group (*p* = 0.16); 23.93 g ±3.49 in the telotristat group (*p* = 0.40); and 23.0 g ± 3.89 in the combination group (*p* = 0.09)] (Figure 4C).

NT-ProBNP levels decreased in a non-significant, but clinically relevant, manner in the three drug-treated groups compared with the control group (Figure 5A). Specifically, NT-ProBNP decreased in the octreotide group by 28.2% (*p* = 0.49), in the telotristat group by 45.9% (*p* = 0.51), and in the combination group by 54.1% (*p* = 0.34) (Figure 4A). In contrast, plasma serotonin levels slightly decreased in a non-significant manner in the telotristat group compared with the control group (20.27%, *p* = 0.06) (Figure 4B). However, no decreases in serotonin levels were observed in the octreotide- or combination-treated groups (Figure 5B).

Finally, fibrosis of the tricuspid valve occurred in all groups. Valve fibrosis decreased in a non-significant manner in the three drug-treated groups. The mean fibrosis value in the control group was 35%, while this percentage was 26.5%, 30%, and 28.8% in the octreotide, telotristat, and combination groups, respectively (Figure 6). Representative images of valve fibrosis in the control, octreotide, telotristat, and combination groups are depicted in Figure 6B–E, respectively. Fibrosis was lower in the octreotide and telotristat groups compared with the control group.

## 3. Discussion

According to current clinical guidelines, surgery is considered the treatment of choice for local or locoregional disease in NENs, especially in grades 1 and 2 tumors [15,16]. Clinical symptoms due to hormone excess should be treated preoperatively. Specifically, radical resection of localized small intestine NENs reduces the risk of bowel obstruction and intestinal ischemia and should be accompanied with systematic mesenteric lymphadenectomy [17]. These tumors tend to present with multifocal lesions and nodal involvement is frequent. When distant metastases are already present, surgery could be considered when large mesenteric masses can cause acute/chronic intestinal obstruction or intestinal ischemia, even if total resection could be achieved in ≤80% of cases in hands of experienced surgeons [18]. Additionally, surgery of liver metastasis could be considered if R0 is expected, debulking surgery for symptoms control could be also considered [19], locoregional therapies could be also considered in cases of liver inoperable metastasis and uncontrolled CS [3].

CHD is a severe complication of CS that increases morbidity and mortality in these patients. Clinically, right heart failure is usually observed; thus, standard medical treatment, and in some cases surgical management, for heart failure is required in these patients [3]. Importantly, valve replacement surgery is indicated only in severe symptomatic CHD with at least 12 months of anticipated post-operative NET-related survival [3]; the selection of the prothesis should be performed individualized [20]. Since CHD can rapidly progress, it should be continuously evaluated in patients with CS or with elevated u-5HIAA [5]. Current management is focused on early detection of CHD, in order to achieve symptoms control and for avoiding additional complications [3]. Ideally, a medical treatment that could prevent carcinoid-related fibrosis, and specifically heart valves fibrosis, would radically change the course and evaluation of patients with CS. In this context, we aimed to develop a mouse model of serotonin-secreting metastatic NEN, in which the specific effect of telotristat on heart valve fibrosis was evaluated. To the best of our knowledge, this is the first report describing the therapeutic actions of telotristat alone or in combination with SSAs in improving heart valve fibrosis using a preclinical a mouse model of CHD.

This study reports the development of a reliable preclinical xenograft mouse model with inoculation of BON1 cells in the spleen for the study of CHD. Specifically, BON1 cell line was derived from a functioning human lymph node metastasis of a pancreatic carcinoid tumor [21,22]. This cell line secretes serotonin, neurotensin and chromogranin A [22,23,24,25]. In this sense, usually xenografts for NENs are developed using subcutaneous injection of the cell line into nude mice [26], wherein these tumors are histologically similar to the original tumor [27], and clinically respond to several drugs and reagents [28]. Therefore, they represent a reliable model for evaluating therapeutic options for primary NENs. Despite this, this subcutaneously-injected xenograft model is unable to develop metastasis, and in consequence, CS related symptoms are not observed [27]. In this context, Jackson and cols. described for the first time a xenograft model of metastatic NEN using BON1 cells in Harlan Sprague Dawley rats after intrasplenic injection [27], this was the reference model for this study. At that time, chromogranin A and serotonin production of the liver metastases was demonstrated, and they also described valvopathy or functional cardiac impairment in most animals [27]. In contrast, our model was performed in nude mice (with inoculation of BON1 cells directly in the spleen), and treatment was started when metastasis and CS had already occurred, assembling the frequent diagnosis of NENs at an advanced stage of disease [4].

The antiproliferative effect of octreotide in NENs has been reported in vitro [23,25], in vivo [27,28], in clinical studies and even in clinical guidelines [29,30,31]. Probably, longer duration of the study would have resulted in more significant effect on tumor growth, since octreotide-LAR reaches a plateau in its concentration at 14 days from injection [3], and mice were treated only for 6 weeks, compared with 12 weeks and daily octreotide administration described in other reports [27]. Additionally, in other in vivo studies [27,28] treatment started after tumor cells injection, while in our study treatment was started only after a clear development of metastatic disease (based on this natural history of metastasis in this xenograft model).

Clinical trials have reported significant improvement in stool movements and flushing in patients treated with telotristat and SSAs [13,14]. Similarly, telotristat and the combination of telotristat with octreotide decreased feces volume in our study. It is remarkable that this reduction was not observed in the group treated with octreotide; again, this fact could be explained by the study duration and the depot preparation of octreotide.

The impact of sarcopenia and malnutrition in NENs has been recently described in some studies [32,33,34], which is associated with clinical outcomes, survival and prognosis [35]. It is reasonable to think that body composition in patients with disease control, might improve or at least remain stable, but in most cases, this situation is not observed [32]. In consequence, regular nutritional evaluation and early treatment might be necessary in several cases [36]. Despite the slight decrease in stools volume and tumor size, we did not observe significant changes in lean or fat mass, suggesting that the tumor itself is able to affect body composition in NEN patients, which is not reversed by systemic treatment, especially in hormone secreting tumors [32,33,34].

The main mediator of carcinoid-related fibrosis is thought to be the amine derivative serotonin. Despite other co-secreted peptide hormones and amines (bradykinins, tachykinins and histamine) have also been related with cutaneous flushing and respiratory complaints [37,38], only serotonin is indirectly measured in the clinical practice as a marker of disease control in patients with CS [3]. Serotonin is synthesized from l-tryptophan, which is an essential amino acid. TPH converts tryptophan to 5-hydroxytryptophan, which is subsequently converted to serotonin, thus, TPH is considered the rate limiting step in the synthesis of serotonin [11]. Regularly, serotonin is metabolized by the liver, but in case of liver metastases, CS- related symptoms occur [39,40]. Exceptionally, CS is observed in patients with bronchial or ovary carcinoids, without liver metastasis [41]. As previously mentioned, CS was confirmed in our model based on the increased volume of feces and the development of heart valve fibrosis in all groups.

In patients, systemic levels of serotonin can be measured by tracking the urinary metabolite u-5HIAA. When elevated u-5HIAA is observed, usually the tumor is widely-spread and associated with severe CS and CHD [13,42]. In this study, u-5HIAA was not determined due to the difficulty in collecting 24-h urine sample. In contrast, plasma serotonin was determined. It has been previously reported that telotristat strongly decreases serotonin secretion in a dose-dependent manner in 2-dimension and 3-dimension in vitro experiments in BON1 cells, even despite this cell line seems to be less sensitive to the effect of this drug than other pancreatic, serotonin secreting NEN cell lines [23]. However, in our study, a slightly non-significant decrease in plasma serotonin was observed in the telotristat group but not in the combined treatment group. It has been previously reported that, in vitro, the combination treatment of telotristat with octreotide is slightly less effective in reducing serotonin levels of BON1 cells than the effect of telotristat alone [23], but this finding has not been reported in clinical trials [13,14]. It is remarkable that in our model serotonin levels did not change in the octreotide group, wherein the study duration might probably have influenced this endpoint. Additionally, plasma serotonin is not the ideal marker for tumor follow-up in patients, since most available measurements employ platelet poor plasma, in which serotonin concentration is low. This fact, together with the sample handling, can easily become a source of errors [43]. For that reason, few reports of circulating serotonin levels in humans are available [5].

Interestingly, our study revealed that plasma NT-proBNP decreased in all groups in a clinically relevant manner. This is a very significant clinical marker, since NT-proBNP is released by heart atria and ventricles in response to the increased wall stress, which is a consequence of volume and/or pressure overload [44,45]. Specifically, NT-proBNP levels positively correlate with the severity of CS, the New York Heart Association (NYHA) functional class, and overall mortality [46,47]. In consequence, currently NT-proBNP is regularly determined in patients for monitoring CHD in combination with echocardiography [3,47]. We did not find in the literature specific reports about changes in NT-proBNP levels in patients treated with telotristat. Ideally, transthoracic echocardiogram should have been performed, but due to the mice size, it was not possible to obtain clear images of the tricuspid and pulmonary valves; thus, we have not reliable results to be reported. These technical limitations were also reported by Jackson and cols [27].

Finally, we observed slight and non-significant reductions in valve fibrosis. Jackson and cols. also reported lower valve abnormalities in rats that received treatment with octreotide [27]. In contrast to our study, a specific percentage of reduction is not reported. Additionally, daily octreotide during 3 months was administered [27]. However, we did not find in the literature any specific reports about valve fibrosis reduction in patients treated with telotristat. Importantly, due to tissue availability, our immunohistochemical results were not molecularly confirmed, but this evaluation could provide confirmatory results in future studies.

Taken together, our results suggest that telotristat might be a valuable medical tool for preventing valve fibrosis is patients with CHD, especially when used in combination with SSAs. These results should be interpreted cautiously, another study including a larger number of animals could be performed, but ideally, a randomized prospective controlled clinical trial should be performed, in which telotristat could be compared with standard of care in the management of CS. Thle trial would be very valuable specially if an earlier use of telotristat is considered in patients with elevated u-5HIAA independently of the presence of uncontrolled diarrhea.

## 4. Materials and Methods

### 4.1. Establishment of the Mouse Model of Serotonin-Secreting NEN

All experimental procedures were approved by the Animal Care and Use Committees of the University of Cordoba and according to the European Regulations for Animal Care. Specifically, 5-week-old ATHYM-Foxn1nu/nu mice (n = 40;) were purchased from Janvier Labs. All mice were bred in-house and maintained under standard conditions of light (12-h light, 12-h dark cycle; lights on at 07:00 h) and temperature (22–24 °C), with free access to tap water and food. They were allowed to acclimate for two weeks. 

Then, a preclinical xenograft mouse model with inoculation of BON-1 cells in the spleen was developed following previously validated protocols [27]. Specifically, the 7-week-old ATHYM-Foxn1nu/nu mice (n = 40) were injected with 1 × 10^6^ BON-1 cells [in 50 μL of PBS] into the spleen by splenectomy as previously reported [48]. BON1 is a pancreatic neuroendocrine cell line that secretes high levels of serotonin additionally with other hormones including neurotensin, pancreastatin and chromogranin A [49]. Briefly, mice were anesthetized with isoflurane and maintained under continuous flow in order to make a small left subcostal flank incision, and for the spleen exteriorization. Tumor cells were injected into the spleen using a 27-gauge needle. The spleen was returned to the abdomen, and the wound was closed in one layer with wound stitches. Painkillers were administered once after surgery. All animals were monitored after this procedure to ensure the individual well-being of each mouse.

### 4.2. Drugs, Reagents, and Treatment Groups

Telotristat was purchased from MedChemTronica (Sweden). Octreotide-LAR was obtained from Novartis Pharma (Basel, Switzerland). Five weeks after BON-1 cells inoculation, four experimental conditions were administered: (1) a vehicle-treated control group (n = 10); (2) a telotristat treated group who received 14 mg/kg/day orally (n = 10); (3) an octreotide treated group that received 10 mg/kg via intramuscular at day 0 and 30 (n = 10); and, (4) a combination treated group that received oral telotristat and intramuscular octreotide. The weight of the mice, cage bed, and waste were monitored twice per week. Sacrifice was carried out either after 6 weeks of treatment or when the condition of an individual mouse was approaching a terminal state (i.e.: significant weight/movility loss), specifically at week 4th in six mices (two in the octreotide, two in telotristat and two in the combination arm) and at 5th week one mice in the control group. Trunk blood was collected in tubes and heart tissue was collected and embedded in 10% formalin paraffin (for the subsequent immunohistochemistry analyses, assessed by an expert pathologist; see below). A schematic overview of the protocol is depicted in Figure 7.

### 4.3. Determination of Whole Body Composition

Whole body composition (fat mass, lean mass, and extracellular water content) was assessed in live mice using a Body Composition Analyzer E26-240-RMT (EchoMRI LLC, Houston, TX, USA) just before sacrifice.

### 4.4. Assessment of Plasma NT-proBNP and Serotonin

Plasma NT-proBNP and serotonin were determined in the collected plasma samples at the time of sacrifice using specific ELISA kits following the manufacturer’s instructions. Specifically, NT-proBNP ELISA was purchased from Cusabio (Houston, TX, USA; lower limit of detection 0.039 ng/mL), while serotonin ELISA was purchased from ALPCO (Salem, NH, USA; limit of detection 6.2 ng/mL; limit of quantification 10.2 ng/mL).

### 4.5. Heart Valve Histology

Heart tissues from each mouse were fixed in 10% formalin, paraffin-embedded, and sectioned in 7 μm sections for hematoxylin-eosin staining following standard protocols, as previously described [50]. After scanning the slides with GTX450 (Leica biosystems) and conducting the initial histologic evaluation, histochemistry with Masson’s trichrome was performed (Ventana system; Roche). Histochemistry was performed following the manufacturer’s instructions. Fibrosis area was determined using ImageJ version 1.54. An expert pathologist performed the histopathological analyses of the heart valves following a blinded protocol.

### 4.6. Statistical Analyses

Continuous variables were expressed as means with standard deviation and categorical variables were described as proportions. Between-group comparisons were analyzed using the Mann–Whitney U test (nonparametric data). Paired analysis was performed using the Wilcoxon test (nonparametric data). Statistical analyses were performed using SPSS statistical software version 20, and GraphPad Prism version 9. *p*-values < 0.05 were considered statistically significant.

## Figures and Tables

**Figure 1 ijms-25-02036-f001:**
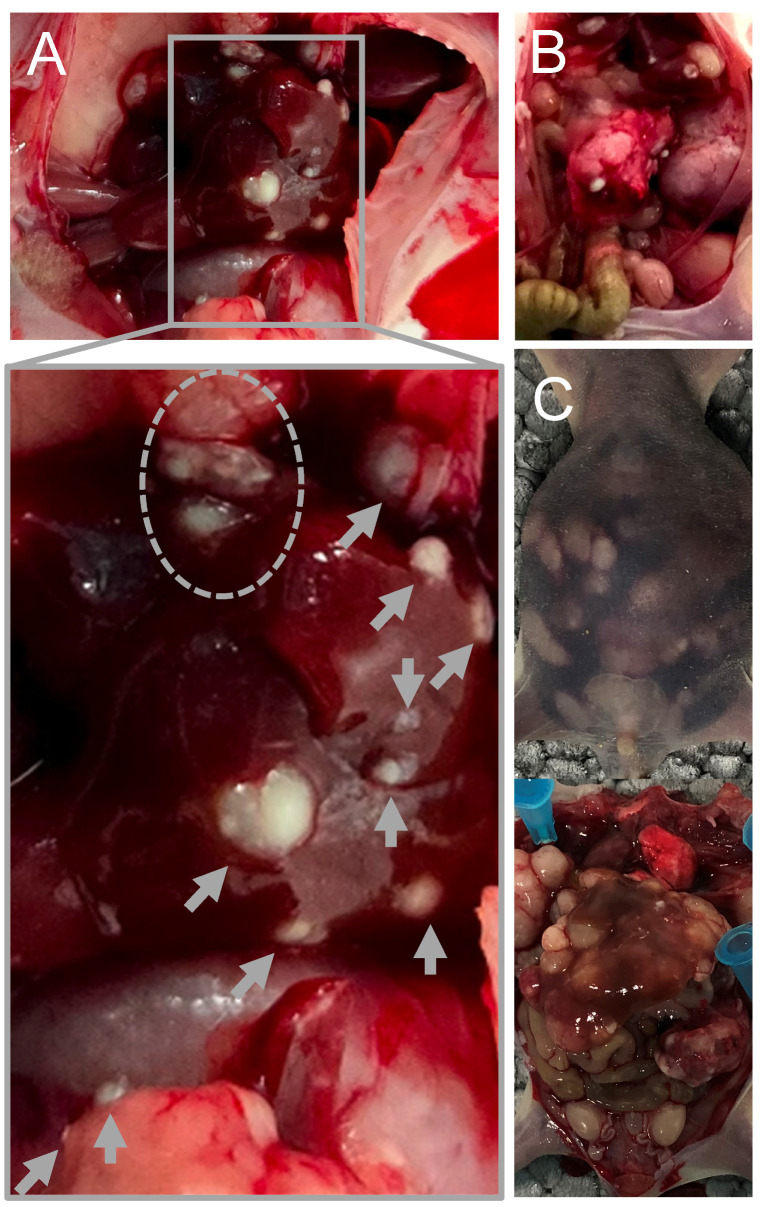
Representative images capturing metastases observed at the terminal stage in the preclinical xenograft mouse model. Panels (**A**–**C**) depict distinct individuals with visually identified metastases. Metastases in the liver is indicated by gray arrows.

**Figure 2 ijms-25-02036-f002:**
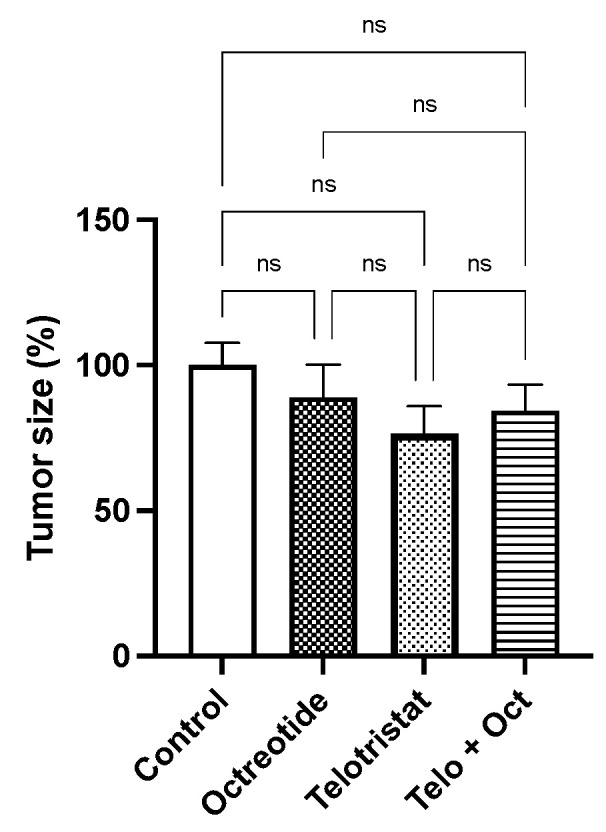
Changes in volume of the primary tumor (expressed in percentage) in mice treated with placebo, monthly octreotide, daily telotristat, or a combination of octreotide and telotristat. Data represent the mean with standard deviation. Legend: ns: non-significant.

**Figure 3 ijms-25-02036-f003:**
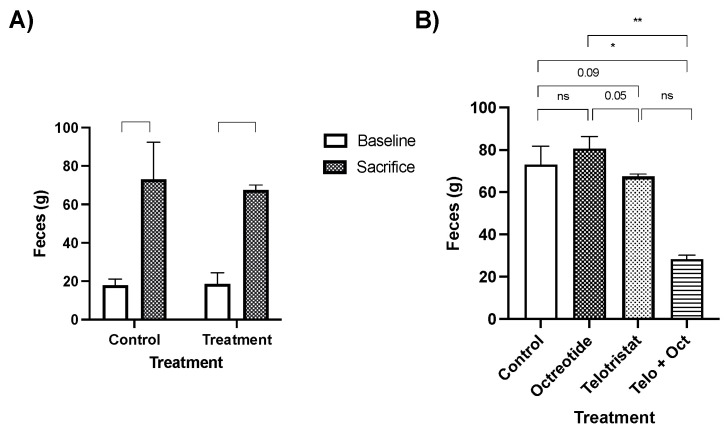
Changes in feces volume (in grams) at baseline and at the end of the study in the control group and the treated mice (**A**). Comparison of feces volume at the beginning and end of the study in the control group and the treated mice (treated with monthly octreotide, daily telotristat, or a combination of octreotide and telotristat). (**B**) Data represent the mean with standard deviation. Legend: *: *p* < 0.05; **: *p* < 0.01; ns: non-significant.

**Figure 4 ijms-25-02036-f004:**
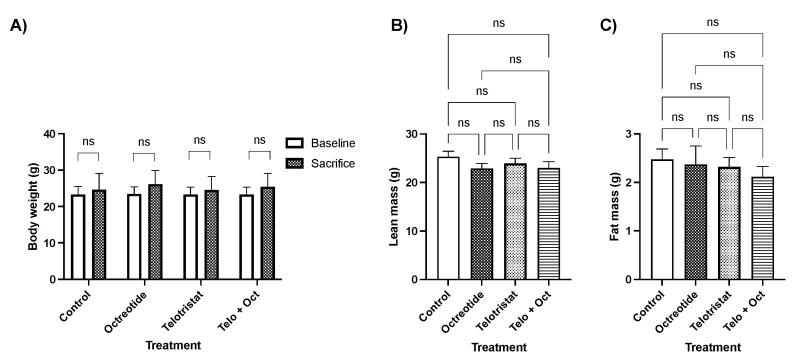
Comparison of body composition measurements between mice treated with either a placebo, monthly octreotide, daily telotristat, or a combination of octreotide and telotristat. (**A**) Changes in body weight at baseline and after sacrifice; (**B**) comparison of lean mass at sacrifice; (**C**) comparison of fat mass at sacrifice. Data represent the mean with standard deviation. ns: non-significant.

**Figure 5 ijms-25-02036-f005:**
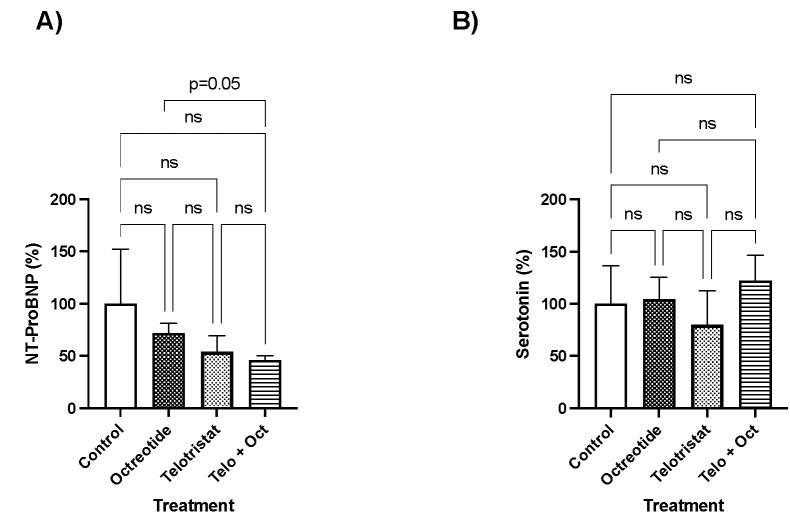
Plasma marker levels at sacrifice. (**A**) NT-proBNP levels as percentages; (**B**) serotonin levels as percentages. Data represent the mean with standard deviation.

**Figure 6 ijms-25-02036-f006:**
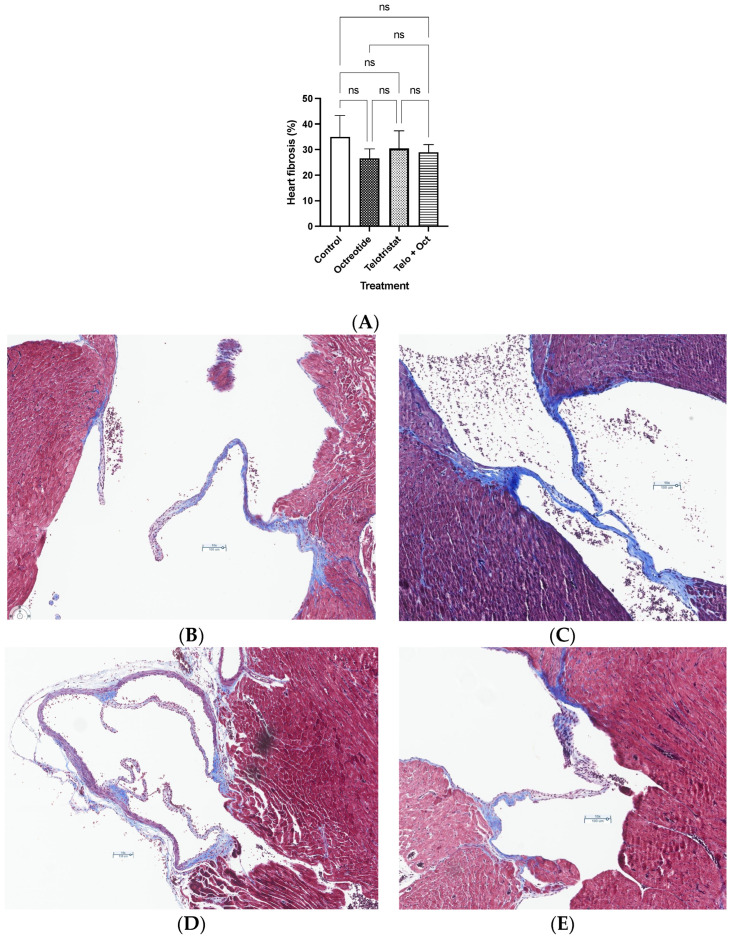
(**A**) Heart fibrosis at sacrifice using Sirius red immunostaining. Data represent the mean with standard deviation. Representative images of valve fibrosis evaluated using histochemistry with Masson’s trichrome using a 10× scale in (**B**) the control group; (**C**) the octreotide group; (**D**) the telotristat group; (**E**) the combination group (telotristat and octreotide).

**Figure 7 ijms-25-02036-f007:**
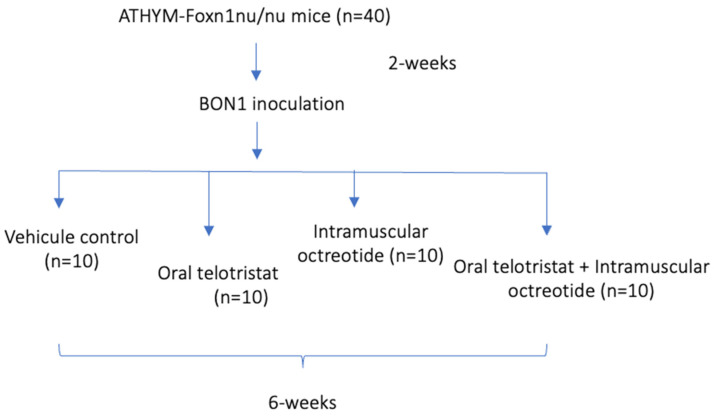
General overview of the study.

## Data Availability

Data contained within the article.

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
