# Peer review of "Does Telotristat Have a Role in Preventing Carcinoid Heart Disease?"

_ijms, 2024, doi:10.3390/ijms25042036_

Round 1

Reviewer 1 Report

Comments and Suggestions for Authors

The article describes a mouse model for CS and the clinical efficacy of Telotristat. The conclusion is supported by the data provided but more results are needed to support the conclusion.

1. The authors mention that IHC was done, please provide relevant images.

2. Cardiac fibrosis was evaluated histologically, please provide H & E and trichrome or pentachrome or staining for elastin images in all groups.

3. The authors mentioned that heart functions were assessed using ECHO, please provide the results. If not done, why (since no data is included)? 3D USG/ECHO findings should be included for concluding that this is a model for CS.

4. line 99- ant should be and.

5. Figure 2 panel A- please include statistics. Also, what treatment the authors are talking about in the X-axis, please mention.

6.significant weight/movility loss). 286 Trunk blood mm was? please check for the completeness of the sentence and spelling.

7. The groups, treatment regimen, doses, and frequency must be described elaborately.

8. H & E staining methods and IHC methodology should be included.

9. The pathological changes at the molecular level must be investigated and the results must be supported by gene and protein analysis (mainly for fibrosis).

Comments on the Quality of English Language

Minor editing is needed.

Author Response

We sincerely thank the Reviewer for the constructive comments, which we found very helpful towards improving the quality of our study. Accordingly, specific changes have been made in the manuscript, based on these comments, as it is described in detail below in a point-by-point description of the changes introduced, and on how Reviewer’s concerns were addressed. Changes in the manuscript are indicated in red

REVIEWER 1

Reviewer: The authors mention that IHC was done, please provide relevant images. Cardiac fibrosis was evaluated histologically, please provide H & E and trichrome or pentachrome or staining for elastin images in all groups.

Authors: Representative images of masson's trichrome in all groups have been included. We realized that the images were not visible in the previous version of our manuscript.

Reviewer: The authors mentioned that heart functions were assessed using ECHO, please provide the results. If not done, why (since no data is included)? 3D USG/ECHO findings should be included for concluding that this is a model for CS.

Authors: In the manuscript  we described that body composition was assessed using a body composition Analyzer E26-240-RMT (EchoMRI LLC). We also performed and echocardiogram, but due to the mice size and probe size, results were not reliable, thus, they were not included in the manuscript. Due to the relevance of this clinical parameter, we mentioned this limitation in the discussion section of the revised manuscript.

Reviewer: line 99- ant should be and.

Authors: This typo has been corrected

Reviewer: Figure 2 panel A- please include statistics. Also, what treatment the authors are talking about in the X-axis, please mention.

Authors: This information has been added in the revised version of our manuscript.

Reviewer: significant weight/movility loss). 286 Trunk blood mm was? please check for the completeness of the sentence and spelling.

Authors: This typo has been corrected

Reviewer: The groups, treatment regimen, doses, and frequency must be described elaborately.

Authors: This information has been added in Materials and Methods section of the revised version of our manuscript.

Reviewer:  H & E staining methods and IHC methodology should be included.

Authors: This information has been added in Materials and Methods section of the revised version of our manuscript.

Reviewer:  The pathological changes at the molecular level must be investigated and the results must be supported by gene and protein analysis (mainly for fibrosis).

Authors: Tissue availability is limited in these models due to the small size of the mice. Based on this (and in the fact that echocardiogram was not reliable), we decided to choose IHC for evaluating heart fibrosis. We have mentioned this limitation in the revised version of our manuscript.

Reviewer 2 Report

Comments and Suggestions for Authors

In this study, the authors tried to generate a mouse model of and carcinoid heart disease

(CHD) to evaluate the effect of telotristat ethyl on right valve fibrosis, and compare its effect not only with placebo, but also with octreotide. The authors suggested that telotristat might be a valuable medical tool for preventing valve fibrosis is patients with CHD, especially in combination with somatostatin analogs (SSAs).

Comments

The reviewer has some concerns as follows:

1.     In the Results, the authors described that “A xenograft model for serotonin secreting NEN was established wherein primary tumor and metastases were observed at sacrifice. Valve fibrosis was also observed in all groups according to the histological analysis.”. However, there are no any data to support that this disease model is successful. The authors should provide these data.

2.     In Figure 2A, the statistically significant symbols are lacking. In B, please explain the numbers for 0.05 and 0.09 in the legend. Moreover, in the legend, the “***p<0.001” seems not to be necessary, because there is no *** in the figure.

3.     In lines 114-115, the units for the data of lean mass are lacking.

4.     In Figure 4A, please explain the numbers for 0.05 in the legend.

5.     In lines 253-254, the authors concluded that “Taken together, our results reveal that telotristat might be a valuable medical tool for preventing valve fibrosis is patients with CHD, especially in combination with SSAs.”. The author over-explains and extrapolates the results of this animal study. This study seems unable to be extrapolated to the actual human situation.

6.     In general, in the present state, the presented results cannot support the conclusions.

Author Response

In this study, the authors tried to generate a mouse model of and carcinoid heart disease

(CHD) to evaluate the effect of telotristat ethyl on right valve fibrosis, and compare its effect not only with placebo, but also with octreotide. The authors suggested that telotristat might be a valuable medical tool for preventing valve fibrosis is patients with CHD, especially in combination with somatostatin analogs (SSAs). 

Comments

The reviewer has some concerns as follows:

Reviewer: In the Results, the authors described that “A xenograft model for serotonin secreting NEN was established wherein primary tumor and metastases were observed at sacrifice. Valve fibrosis was also observed in all groups according to the histological analysis.”. However, there are no any data to support that this disease model is successful. The authors should provide these data.

Authors: we agree with the reviewer concerning this point. A supp figure that demonstrates the presence of liver metastasis was added. The representative images of heart fibrosis were already included in Figure 5, but we noticed that it was not possible to visualize them.

Reviewer: In Figure 2A, the statistically significant symbols are lacking. In B, please explain the numbers for 0.05 and 0.09 in the legend. Moreover, in the legend, the “***p<0.001” seems not to be necessary, because there is no *** in the figure. 

Authors: In the revised version of our manuscript, legend was modified, the meaning of *, **, ***, p.05 and 0.09 have been added.

Reviewer: In lines 114-115, the units for the data of lean mass are lacking.

Authors: This mistake was corrected in the revised version of our manuscript.

Reviewer: In Figure 4A, please explain the numbers for 0.05 in the legend.

Authors: The explanation was added in the revised version of our manuscript

Reviewer: In lines 253-254, the authors concluded that “Taken together, our results reveal that telotristat might be a valuable medical tool for preventing valve fibrosis is patients with CHD, especially in combination with SSAs.”. The author over-explains and extrapolates the results of this animal study. This study seems unable to be extrapolated to the actual human situation.

Authors: we agree with the reviewer regarding this point, thus, the conclusions have been reorganized in the revised version of our manuscript.

Reviewer 3 Report

Comments and Suggestions for Authors

Thank you for permitting me to review this manuscript

here are my queries  

what is the incidence of heart failure in this category of patients , 

what is the surgical indication of tumor removal 

what percentage of patients  are totally controlled by octreotid and other medication , without needing surgery 

Please draw a flowchart for the experiment 

Line 287 please correct , something is missing , and there is red ink 

how many mouse were sacified  for intolerance before  6 weeks ? 

Please  be more cautious for the conclusion since the result are not very conclusive and it is an animal study

Author Response

We sincerely thank the Reviewer for the constructive comments, which we found very helpful towards improving the quality of our study. Accordingly, specific changes have been made in the manuscript, based on these comments, as it is described in detail below in a point-by-point description of the changes introduced, and on how Reviewer’s concerns were addressed. Changes in the manuscript are indicated in red

Reviewer: what is the incidence of heart failure in this category of patients

Authors: It affects 50% of patients with CS. This information has added in the introduction of the revised version of our manuscript.

Reviewer: what is the surgical indication of tumor removal 

Authors: This information has been added to the introduction of the revised version of our manuscript.

Reviewer: what percentage of patients are totally controlled by octreotide and other medication, without needing surgery 

 Authors: Information about medical and surgical treatment in small intestine NENs has been included in the discussion of the revised version of our manuscript. Additionally, information about valve surgery was included.

Reviewer: Please draw a flowchart for the experiment 

 Authors: As suggested, a flowchart has been included in the Supp Figure 2

Reviewer: Line 287 please correct , something is missing , and there is red ink 

Authors: This mistake has been corrected.

Reviewer: how many mouse were sacrified  for intolerance before  6 weeks ? 

Authors: This information has been included in the revised version of our manuscript

Reviewer: Please  be more cautious for the conclusion since the result are not very conclusive and it is an animal study

Authors: we agree with the reviewer regarding this point, thus, the conclusions have been reorganized in the revised version of our manuscript.

Round 2

Reviewer 1 Report

Comments and Suggestions for Authors

None

Reviewer 2 Report

Comments and Suggestions for Authors

This revised manuscript has a great improvement and can be accepted.